# Graphitized Carbon Xerogels for Lithium-Ion Batteries

**DOI:** 10.3390/ma13010119

**Published:** 2019-12-26

**Authors:** Maria Canal-Rodríguez, Ana Arenillas, Sara F. Villanueva, Miguel A. Montes-Morán, J. Angel Menénedez

**Affiliations:** Instituto Nacional del Carbón (INCAR-CSIC), Francisco Pintado Fe 26, 33011 Oviedo, Asturias, Spains.villanueva@incar.csic.es (S.F.V.); miguel@incar.csic.es (M.A.M.-M.)

**Keywords:** carbon xerogels, graphitization, graphene, lithium-ion batteries

## Abstract

Carbon xerogels with different macropore sizes and degrees of graphitization were evaluated as electrodes in lithium-ion batteries. It was found that pore structure of the xerogels has a marked effect on the degree of graphitization of the final carbons. Moreover, the incorporation of graphene oxide to the polymeric structure of the carbon xerogels also leads to a change in their carbonaceous structure and to a remarkable increase in the graphitic phase of the samples studied. The sample with the highest degree of graphitization (i.e., hybrid graphene-carbon xerogel) displayed the highest capacity and stability over 100 cycles, with values even higher than those of the commercial graphite SLP50 used as reference.

## 1. Introduction

The current increase in the environmental awareness of society is leading to a reduction of the use of fossil fuels as an energy source. The growing concern about fossil fuels, is attributed to their non-renewable nature and to the environmental pollution related with their use. One possible solution to tackle fossil fuel dependence is to use renewable sources—such as solar, eolic, or biomass—to generate electrical energy. However, these natural sources have the disadvantages of being intermittent and very dependent on climate factors. Therefore, to expand these clean energy sources it is necessary to develop advanced energy devices able to store large amounts of energy as it is produced so that it can be supplied when it is needed (i.e., grid regulation). For this purpose, lithium-ion batteries (LIBs) are the most widely used devices due to their high capacity. Nowadays, they are also commonly employed in most mobile electronic devices (laptops, tablets, smartphones, etc.). Special attention is also being focused on their use in electric vehicles where highly-efficient electrical energy storage systems are required [1,2]. 

The operating mechanism of lithium-ion batteries is based on the transformation of chemical energy into electrical energy [3,4], and are composed by electrochemical units known as cells. Each cell is made up of two electrodes separated by a porous fiberglass and submerged in an electrolyte [3,5,6]. The electrolytes employed are commonly composed of lithium salts dissolved in an organic solvent, while the cathode and the anode are made up of different transition metals–lithium oxides, and carbonaceous materials, respectively [4,7,8,9]. Its working principle is based on the extraction of lithium ions from the cathode, that migrate through the electrolyte to be intercalated into the structure of the carbon material in the anode [6,10,11].

Given that the performance of the LIBs is mainly controlled by the electrode materials, its suitable selection is a critical factor for ensuring a good workability of the device.

For the anode, graphite is the carbon most commonly used as active material in commercial LIBs since it is easily accessible, cheap, and offers operational advantages over other non-carbon materials [1,12]. However, its maximum specific capacity is limited to 372 mAh g^−1^ (1 lithium atom per 6 carbon atoms) and its structure is prone to irreversible expansion due to the continuous insertion-release of ions, which causes degradation of the electrode and therefore short durability of the cell [3,4,11,13]. For this reason, other carbonaceous materials, including carbon nanotubes, carbon nanofibers or graphene, appear as possible substitutes for graphite [1,14,15].

The studies in which graphite has been replaced by other less ordered carbons show higher values of capacity, which indicates that the storage of lithium is not only dependent on its intercalation between graphitic layers, but also on other mechanisms of charge storage. These alternative mechanisms have not yet been brought to light, although some researchers consider that defects in the turbostratic structures could be acting as active sites that adsorb the Li^+^ ions [16,17]. 

Consequently, carbon xerogels would seem to be good candidates for use as electrodes in LIBs due to the fact that their structural and chemical properties can be custom-made for a specific application [18,19,20]. Carbon xerogels are made up of nodules interconnected with each other in a three-dimensional network. The voids generated between the nodules are the macro (>50 nm) or mesopores (2–50 nm) depending on their diameters. While inside the nodules there are also pores with diameters below 2 nm known as micropores. The volume and pore size distribution can be fitted just changing their synthesis conditions [21,22,23,24,25,26,27,28]. This high volume of pores makes xerogels to be carbons that have a lot of defects and it is this that causes the electrons to move more slowly than in other more ordered materials resulting in a lower electrical conductivity which is one of the essential requirements of electrochemical devices. To overcome this drawback, some authors have introduced a highly conductive additive into the xerogel structure to facilitate a better electron mobility [25,26,27,28,29,30,31,32,33,34,35] and to obtain a better performance as electrode in supercapacitors, while others have used graphitization processes at high temperatures (2000–3000 °C) to obtain highly ordered carbon xerogels with enhanced conductivities [33,36,37,38]. Each different approach gives rise to final carbons that combine ordered with turbostratic parts in a single material, resulting in diverse chemical properties, all of which contribute in different ways to the energy storage mechanism. 

The main goal of this study is to analyze how the different structural and porous properties of carbon xerogels affect the workability of lithium-ion batteries, and to discover the most suitable material to improve this performance. To this end, the two alternatives mentioned above for obtaining xerogels with better electrical conductivities (i.e., heat treatments and the addition of conductive compounds) were applied. Thus two carbon xerogels with different pore sizes and their respective hybrid graphene-carbon xerogels were synthesized and subsequently treated at high temperature. Samples were then tested as electrodes in lithium-ion batteries and their performance was analyzed in terms of battery reversible capacity, irreversible capacity in the first cycle, and retention capacity along cycling.

## 2. Materials and Methods 

### 2.1. Synthesis and Graphitization of the Carbon Xerogels Studied

Two organic xerogels were synthesized using resorcinol (R) and formaldehyde (F) as reactants, deionized water as solvent, and a solution of NaOH as basification agent. Firstly, the resorcinol was dissolved in the water and subjected to magnetic stirring until total dissolution. Subsequently, the formaldehyde was added and the mixture was also stirred to obtain a homogeneous solution. Finally, the pH of the precursor solution was adjusted before being placed in a microwave oven for 5 h at 85 °C, where gelation, curing and drying of the xerogel took place [39,40,41,42]. The initial conditions applied in terms of dilution (D), R/F ratio and pH were as follows: (i) D = 9.7, R/F = 0.5 and pH = 5.8, (ii) D = 7.8, R/F = 0.3 and pH = 4.7. The organic xerogels thus synthesized exhibited different porous characteristics with diameters of ca. 100 and 300 nm, respectively.

Two graphene oxide (GO) hybrid organic xerogels were also synthesized following the same procedure, but replacing the water used as solvent by a 0.5 wt% GO suspension in water [32,34,35]. The rest of synthesis conditions were therefore the same as those used in the case of the pristine xerogels.

The GO suspension was produced in accordance with the modified Hummers–Offeman method using natural graphite (NG) flakes BNB90 purchased from Imerys Graphite and Carbon (Bodio, Switzerland), by liquid phase exfoliation through the ultrasonication of graphite oxide [43]. The synthesis process has been reported in detail in previous studies [35].

The organic xerogels were subjected to different high temperature treatments. The samples were carbonized in a vertical oven, under a N_2_ flow of 300 mL min^−1^, at a heating rate of 50 °C min^−1^ up to 1000 °C. The samples were maintained at this temperature for 2 h. With this treatment, the organic xerogels released the labile surface groups as volatiles, obtaining their respective carbon xerogels with mainly condensed aromatic rings in their structure. Consequently, during this heating process, the GO used as an additive in some of the samples, and incorporated within the polymeric structure during the synthesis, was also reduced by the effect of temperature and the evolution of the oxygen functionalities. As a result of the heat treatment, these samples will henceforth be referred to as graphene-carbon xerogels. 

These carbonized samples were subsequently treated at a higher temperature (i.e., 2800 °C) in a graphitization oven (Xerion Advanced Heating, Germany), under an Ar flow of 2 L min^−1^. The heating rates selected were: (a) 50 °C min^−1^ up to 700 °C, (b) 100 °C min^−1^ from 700 to 1000 °C, (c) 25 °C min^−1^ from 1000 to 2000 °C, and (d) 10 °C min^−1^ in the range of 2000–2800 °C. The residence time at the maximum temperature was 2 h.

The samples studied were labelled CX or GX in reference to the heat treatment performed (i.e., carbonization at 1000 °C or graphitization at 2800 °C, respectively), followed by mean pore size and of the suffix ‘GO’ in the case of the hybrid graphene-carbon xerogels. For example, sample CX-100 was obtained by the carbonization at 1000 °C of a RF xerogel with a mean pore size of 100 nm, whilst GX-300GO was obtained by the graphitization at 2800 °C of the xerogel resulting from a precursor mixture to obtain a mean pore size of 300 nm but using a GO suspension instead of water as a solvent. A commercial graphite (TIMREX^®^ SLP50) was also employed for comparative purposes.

### 2.2. Structural and Porous Characterization

X-ray diffraction (XRD) results were collected on a Bruker D8 Advance powder diffractometer equipped with an internal standard of silicon powder and a monochromatic Cu-KαX-ray source with 0.1542 nm of wavelength. Data was collected over 2θ range from 5° to 80° with a step size of 0.02° and an interval of 3 s per step. Since the xerogels studied are not purely graphitic carbons, the (002) reflection was deconvoluted in two Gaussian curves corresponding to both amorphous and graphitic contributions [44]. The % of graphitic contribution was obtained by the ratio between the area corresponding to this graphitic component and the total area of the (002) peak. The interplanar distance (d002) and the crystallite size perpendicular to the basal plane (Lc) were calculated according to the Bragg’s Law and Scherrer´s equation, respectively.

The porous properties of the samples were evaluated on the basis of N_2_ adsorption-desorption isotherms at −196 °C (Micromeritics Tristar 3020), and by means of mercury porosimetry (Micromeritics Autopore IV). The S_BET_ and V_micro_, were determined by applying the Brunauer–Emmett–Teller (BET) and Dubinin–Raduskevich (DR) equations to the adsorption branch of the nitrogen isotherms. The total pore volume (Vp) was determined from the amount of nitrogen adsorbed at the saturation point (p/p0 = 0.99). As the nitrogen adsorption technique is not precise enough for samples with macropores, mercury porosimetry analysis was also performed. The conditions chosen were a pressure of up to 228 MPa, a surface tension of 0.485 N m^−1^ and a contact angle of 130°. The mesopores volume (V_meso_), macropores volume (V_macro_) and average pore size (D_p_) were estimated applying Washburn’s intrusion theory.

High resolution electron transmission microscopy (HRTEM) was also used to examine the samples. In a previous step, the samples were dispersed in ethanol, sonicated, sprayed onto a carbon-coated copper grid, and then allowed to air-dry. The analysis was performed in a JEOL JEM-2100F microscope operated at an accelerating voltage of 200 kV. The microscope was equipped with a field emission gun and an ultra-high resolution pole-piece that provided a point-resolution higher than 0.19 nm. The morphology of the samples was also examined using a Quanta FEG 650 scanning electron microscope (SEM). Samples were previously attached to an aluminum tap using conductive double-sided adhesive tape. An accelerating voltage of 25 kV and a secondary electron detector EDT (EverharteThornley) were employed.

To determine the electrical conductivity of the samples, disc-shapes pellets were prepared by mixing 90 wt% of each material with 10 wt% of polytetrafluoroethylene (PTFE) as binder. A detailed description of their assembly is available in the literature [40,45]. The pellets were placed on a chuck made of teflon where its electrical resistivity was evaluated in accordance with the four-point probe technique (FPP). This technique involves placing the four points in contact with the surface of the disc-shape electrodes and applying a constant current in the 9–10 mA range between the four points (DC current source model 6220, Keithley). The corresponding drop in voltage was measured using a digital nanovoltimeter (model 2182 A, Keithley). 

### 2.3. Electrochemical Characterization

To perform the electrochemical characterization, electrodes were prepared by mixing 92 wt% of carbon xerogel, 8 wt% of polyvinylidene fluoride (PVDF) as binder and N-methyl-2-pyrrolidone (NMP). The mixture was kept under stirring to obtain a homogeneous slurry and then deposited by dripping onto pressed copper disks, with a diameter of 10 mm and a thickness between 105–110 µm, subsequently, they were left to dry overnight at 60 °C [33]. After this process, electrodes with an active material thickness of 35–40 µm and a mass loading in the range of 1.3–3.5 mg cm^−2^ were obtained.

The assembly used for the characterization was a two-electrode testing half-cell (Teflon Swagelok^®^), in which the carbon material acted as the cathode, and a Li-metal disk (Aldrich) served as the anode. The electrodes were separated by a 400 µm thick fiber glass separator (Whatmann^®^), onto which several drops of electrolyte (1M LiPF6 in ethylene carbonate (EC): dimethyl carbonate (DMC) 50:50, *v*/*v*) were added. To evaluate the behavior of the materials as electrodes, different tests were performed: (i) charge-discharge (C-D) measurements at C/5 between 0.005 and 1.5 V (vs. Li^+^/Li), (ii) C-D experiments in the same range of voltage but varying the C-rate from C/5 to 10C, i.e. modifying the rate at which the battery is discharged taking into account the theoretical capacity of graphite (372 mAh g^−1^). The equipment used was a VMP3 potentiostat/galvanostat from Biologic.

The irreversible capacity (Cirr), the cell retention (R), and the Coulombic efficiency were calculated by applying the expressions [1]
C_irr_ (%) =(C_dis_ (1st cycle)-C_charge_ (1st cycle))/(C_dis_ (1st cycle)) × 100(1)
R (%) = (C_dis_ (50th or 100th cycle))/(C_dis_ (2nd cycle)) × 100(2)
E_c,cycle i_ (%) = (C_charge_ (cycle i)/ C _dis_ (cycle i)) × 100(3)
where C_disc_ is the discharge capacity (mAh g^−1^) and C_charge_ is the charge capacity (mAh g^−1^).

## 3. Results and Discussion

### 3.1. Structural and Chemical Characterization

To evaluate the contribution of: (i) the pore size of the samples and (ii) the addition of graphene oxide, to the degree of graphitization of the carbon xerogels, XRD analyses were performed.

As can be observed in Figure 1, CX-100 is an almost amorphous material as none of the characteristic peaks of graphite appear in its pattern. Instead, two broad peaks are located at ca. 25° and 45° 2θ [33,46]. In general, the graphitization process leads to the narrowing and the displacement of the maximum of the peak at around 2θ = 25° attributed to the (002) diffraction peak of the graphitic framework. However, the intensity of that peak in the graphitized samples is still very low when compared to that of the SLP50 graphite (Appendix A). Moreover, the graphitic material used as reference also shows signals at ca. 2θ = 43°, 44°, and 54° assigned to the (100), (101), and (004) diffractions, while only a wide peak at ca. 43° 2θ, attributed to the two previous (10) reflections, is observed in the XRD patterns of the graphitized pristine xerogels (i.e., GX-100 and GX-300). That peak seems to unfold into the two contributions (i.e., (100) and (101)) when adding GO, which indicates that the hybrid samples have a more developed three-dimensional structure. These results indicate that graphitized carbon xerogels contain both amorphous and crystalline (more ordered) structures. 

In light of these results, structural parameters were determined considering that the shape of the (002) peak of the patterns of the GXs was a contribution of non-graphitic and graphitic phases [44]. Table 1 shows that the ordered part of both of the graphitized carbon xerogels (GX-300 and GX-100) present values of d_002_ that are close to that of graphite (i.e., 0.335 nm at 2θ = 26.6°). Moreover, their crystalline dimensions along the c-axis (Lc) are larger than those reported for carbon xerogels in the literature [33] although much smaller than that of the reference graphite. In any case, there is a relevant difference between these two GXs, i.e., that the graphitic contribution of sample GX-300 (22.6%) is three times greater than that of GX-100 (7.5%), indicating that the pore size of the carbon xerogels exercises a marked influence on the graphitization of these samples. 

In the case of the hybrid materials, it can be seen that the addition of 0.2 wt% of GO to the precursor solution enhances the graphitic contribution of both of the carbon xerogels, and that their d_002_ and Lc values remain as good as those of the pristine samples (Table 1). Therefore, the incorporation of graphene to the carbon xerogel structure does not contribute to increase the quality of the graphitic structures but their relative abundance. In a previous study [35], it was found that the addition of at least a 0.2 wt% of graphene oxide generates a carbon xerogel made up of interconnected graphene layers embedded into the smaller nano-sheets that conform the carbon xerogel structure. During the graphitization treatment, these interconnected graphene layers may act as a seed that promotes the graphitization process. As a consequence of adding GO to the carbon xerogels, the resulting hybrid GX-GOs consist of a more laminar rather than nodular structures characteristic of the conventional GXs (Appendix A).

The structural examination using HRTEM (Figure 2) supports the results obtained by XRD analysis.

As can be seen in Figure 2, the framework of the graphitic xerogels is formed by two different structures (i) disordered and rolled up planes that correspond to the non-graphitized structure of the gel and (ii) graphitic nanocrystals. Although these two sorts of ordering are easily distinguished in all the materials studied, GX-100 seems the sample with the less defined ordered structure while GX-300 presents more and larger graphitic areas combined with turbostratic zones. In the case of the hybrid gels, both non-graphitic and graphitic structures are again identified in the TEM images. Graphitic areas in this case correspond not only to the xerogel, but also to the graphene embedded into its structure. 

As observed in Figure 3a, where the isotherms of CX-100 and GX-100 are compared, the graphitization process produces a decrease in the volume of micropores, which in turn leads to a decrease of 88% in the S_BET_ of the graphitized sample (Table 1), while the macroporosity remains unchanged (Figure 3b). It would appear, therefore, that graphitization reorganizes the basal planes inside the nodules of the polymeric structure of the xerogels producing a collapse of the micropores [33], but the heat treatment does not modify the size of the nodules and the space between them (i.e., the size of the macropores). 

Carbon xerogels are made up of clusters or nodules that have micropores (<2 nm) inside them. These nodules are interconnected but with certain spaces between them that form the meso (2–50 nm) or macroporosity (>50 nm) [47,48]. According to the literature [49], larger nodules imply xerogels with a larger pore size. As can be seen from the SEM images (Appendix A), GX-300 is made up of bigger nodules, which explains the graphitization of larger volumes of carbonaceous structure.

Another factor to be taken into account is that both of the hybrid carbon xerogels have a wider pore size than their counterparts without graphene (Figure 3b,d), with a pore size of ca. 1000 nm in both cases. This increase in pore size is a consequence of the thermal treatment performed on these samples, which is perceptible even during the carbonization process. The increase in temperature during the carbonization process at 1000 °C causes the oxygen-containing functional groups to decompose into gases that generate a great pressure between the stacked layers of graphene oxide [50]. The literature reports pressures of 40 and 130 MPa being generated at 300 and 1000 °C, respectively [50,51]. This process causes the exfoliation of the GO sheets and an expansion of the hybrid graphene-carbon xerogel structure, as a result of which larger pores are generated (ca. 1000 nm). These pores then remain unchanged during the graphitization process, as can be seen in Appendix A.

The materials containing highly ordered sheets, like those of graphite or graphene, permit electrons to move easily through their structures, as evidenced by the high values of electrical conductivity. On the other hand, disordered structures entail voids and defects that obstruct electron mobility. Besides, usually high porosities also contribute to these discontinuities and the decrease of the electrical conductivity. 

As expected, Figure 4 shows that CX-100 is the sample with the lowest electrical conductivity value (200 S m^−1^), whereas after graphitization the electrical conductivity increased by as much as 674% (K(GX-100) = 1548 S m^−1^). GX-300 displayed the highest electrical conductivity of all the samples. Its value is double that of GX-100 and can be attributed to its greater degree of graphitization, as shown in Table 1. However, despite the fact that the hybrid xerogels are the materials with the highest graphitic contribution (Table 1), their electrical conductivities are not as good as that of GX-300. The expansion of the GO during the carbonization process led to a widening of the carbon xerogels’ porosity. The SEM images (Appendix A) show that in the hybrid xerogels the nodular structure almost disappeared, giving rise to a different carbonaceous framework. As a consequence of this expansion, although ordered structures are predominant in these samples, they may not be well interconnected due to the different carbonaceous structure (i.e., poorly connected polymeric clusters) and the presence of large voids (ca. 1000 nm), that would hamper a good movement of the electrons, leading to a clear decrease in the electric conductivities of the hybrid graphene–carbon xerogel samples. 

### 3.2. Electrochemical Characterization

The samples (except GX-100GO, which has similar chemical and structural properties to GX-300GO) were evaluated electrochemically as electrodes in lithium-ion batteries by means of charge-discharge tests at a rate of C/5 between 0.005 and 1.5 V (vs. Li^+^/Li) (Figure 5).

It should be noted that in the present study a half-cell configuration was used. In an assembly of this type, the insertion of lithium ions inside the carbon structure corresponds to the discharge, whereas the extraction of the ions occurs during the charge.

In Figure 5, during the first discharge, a plateau at around 0.8 V, corresponding to the formation of the solid electrolyte interface (SEI), can be observed in all samples. This interface generates an irreversible consumption of lithium ions. However, the formation of the interface is necessary to prevent the electrode from degrading [17] The dimensions of the SEI interface depend on the surface characteristics of the carbon used as electrode [1,52]. 

In general, the intercalation of the lithium ions within the graphitic layers begins below 0.2 V and is reflected in the form of potential plateaus that appear on the curves. Consequently, carbons with a low degree of graphitization exhibit profiles without any potential plateau [17,46]. In the case of the samples analyzed in this study, a plateau at ca. 0.1 V is clearly visible in the more graphitized sample (i.e., GX-300GO), whilst a continuously decreasing discharge profile is observed in the case of the more amorphous sample (i.e., CX-100). 

The plateau at 0.1 V is also observed in the first charge of the samples, indicating that the intercalation of the Li+ ions is a reversible process. Furthermore, none of the charge profiles show any plateau at 0.8 V due to the irreversibility of the SEI formation.

Coming back to the discharge, although CX-100 shows a gradual decay in the voltage owing to its mainly non-graphitic nature (Figure 1), it is nevertheless the sample that registered the highest capacity. This high lithium storage capacity could be assigned to a greater quantity of active sites, such as pores or edge-type sites, into which the ions can be adsorbed [16]. On the other hand, GX-300GO, which is the sample with the highest graphitic contribution (Table 1), shows the lowest capacity (Figure 5a), while GX-100 and GX-300 present intermediate values. These results could lead to the conclusion that during the first discharge the active sites make a bigger contribution to the capacity than the graphitic planes. Consequently, a greater ratio non-graphitic/graphitic phase, gives rise to an improvement of the charge stored during the first discharge.

Nevertheless, higher non-graphitic/graphitic ratios result in greater irreversible capacity and lower Coulombic efficiencies, as shown in Table 2 (i.e., 72–74% for samples CX-100, GX-100, and GX-300), whereas GX-300GO retains almost half of the charge stored during the first discharge (48.6%) and SLP50 keeps the 89% of its initial capacity (Table 2). The loss of capacity during the first cycle is mostly associated with the SEI formation [33]. Disordered structures present more defects into which the lithium ions can be irreversibly adsorbed.

In cycle 50, GX-300GO displayed the best cell performance with a retention after the SEI formation of an 88% (Table 2) and recording a capacity of 185 mAh g^−1^ (Figure 5b), which is 105 and 33% higher than the capacity of the other graphitic xerogels and SLP50, respectively. In contrast, CX-100 registered the worst capacity (84 mAh g^−1^) with a retention of only a 37% (Table 2). Once again, this loss of capacity can be attributed to its low graphitized nature which on the one hand causes the irreversible storage of lithium ions and on the other hand impedes the rapid movement of ions because of its poor electrical conductivity (Figure 4). The cyclability test results, shown in Figure 6, support these findings.

On the one hand, GX-100 and GX-300 show good stabilities with capacity drops after 100 cycles of just 37 and 30% (Table 2), respectively. However, in spite of the fact that GX-300 shows a greater electrical conductivity and a lower percentage of disordered phase, its capacity is only 9.5% better than that of GX-100. This suggests that although electrical conductivity is important, it is not enough to cause any improvement in cell stability. 

On the other hand, GX-300GO registered the greatest capacity after cycle 10, and showed excellent stability after 100 cycles with a retention of ca. 78% (Table 2). This improvement with respect to the other samples could be attributed to its adequate structure with 30% of graphitic phase (Table 1), whereas in the case of GX-100 and GX-300, their more amorphous structure increased the irreversible storage of Li^+^ ions and reduced capacity after several cycles.

In addition, the performance of GX-300GO was also greater than that of the reference material (SLP50) during all the charge–discharge experiments, with capacity enhancements of around 49% during the first discharge and as much as 60% after the 100th cycle. The higher stability of this xerogel could be attributed to its lowest degradation during cycling. 

One of the main drawbacks of graphite when used as electrodes in batteries is the irreversible modification of their ordered frameworks on account of the expansion of the graphitic sheets during the insertion-extraction of the Li^+^ ions. Unlike the reference material (Appendix A), GX-300GO is a carbon with a large volume of pores (Figure 3d), that may absorb this expansion and thereby contribute to the maintenance of the initial structure. What is more, GX-300GO is not only more stable than SLP50, but also more stable than the rest of the graphitic gels (GX-100 and GX-300) (Table 2), which confirms that the porosity of the carbon xerogels contributes positively to the cyclability of the cells. 

To gain further insight into the electrochemical performance of the samples, different C-rates were applied. The C-rate was increased from C/5 to 10C (Figure 7).

In general, all the samples showed a good stability against the change in C-rate and they managed to recover their initial capacity. However, the sample with the highest degree of graphitization (GX-300GO) attained the highest values at all working conditions. What is more, from C/5 to C (i.e., the values usually required for consumer electronic applications [1]) the sample retained almost 100% of the stored capacity (ca. 160 mAh g^−1^). This is a remarkable result as normally carbon materials with a develop porosity store high capacitances at low C-rates [53,54] but they are unable to retain such a high values when the working conditions become more severe, reaching a loss of capacitance of a 42% when the C-rate goes from 0.1C to 1C [54].

## 4. Conclusions

In this study, carbon xerogels with different pore sizes were subjected to a graphitization process. It was observed that a large nodule size and thus pore size resulted in carbon xerogels with more graphitic structures. On the other hand, the incorporation of graphene oxide into the precursor solution of the xerogels, enhanced the graphitic character of the final hybrid sample considerably. The reason for this is that graphene may act as a seed during the graphitization process. 

When these materials are used as electrodes in Li-batteries, it was observed that the turbostratic part just contributes to improve the capacity of the battery during the first cycle, while the graphitic phase develops the retention capacity of the cell. Moreover, the presence of macropores prevents the electrode from being degraded by the expansion caused by the continuous insertion–disinsertion of lithium ions over cycling. It is for this reason that, the graphitized hybrid graphene–carbon xerogel (GX-300GO) with a high graphitic structure and large pores combines a high capacity and high stability even after 100 cycles, with values higher than those of the commercial graphite used as reference (SLP50).

## Figures and Tables

**Figure 1 materials-13-00119-f001:**
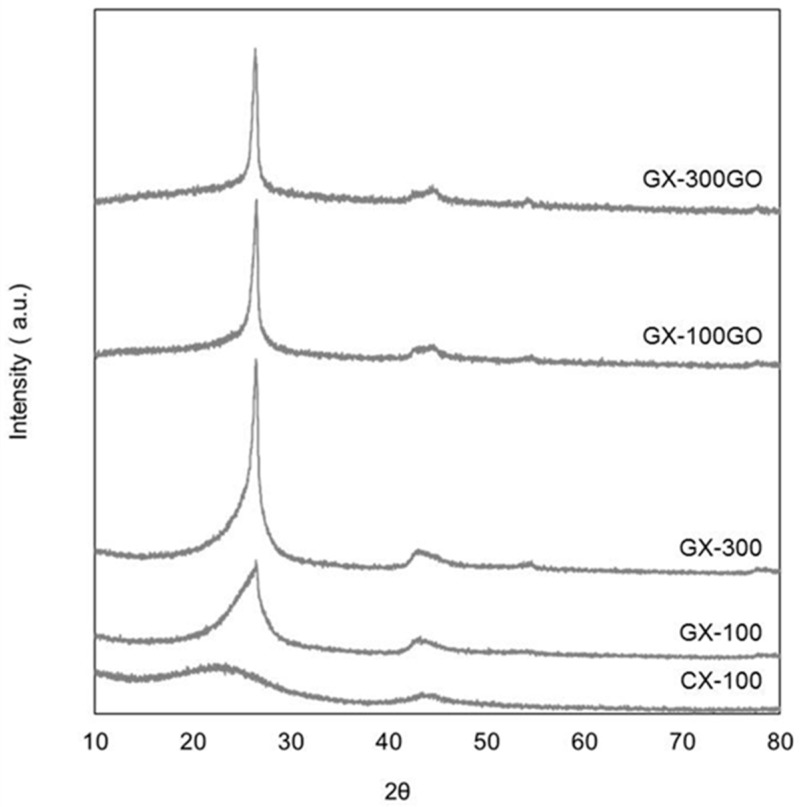
XRD patterns of the xerogels studied.

**Figure 2 materials-13-00119-f002:**
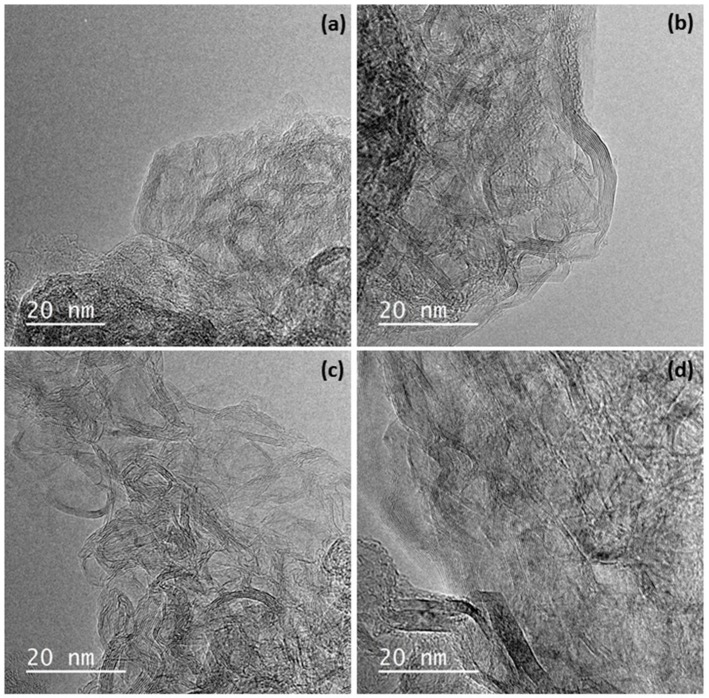
HRTEM images of samples: (**a**) GX-100, (**b**) GX-300, (**c**) GX-100 GO, and (**d**) GX-300 GO.

**Figure 3 materials-13-00119-f003:**
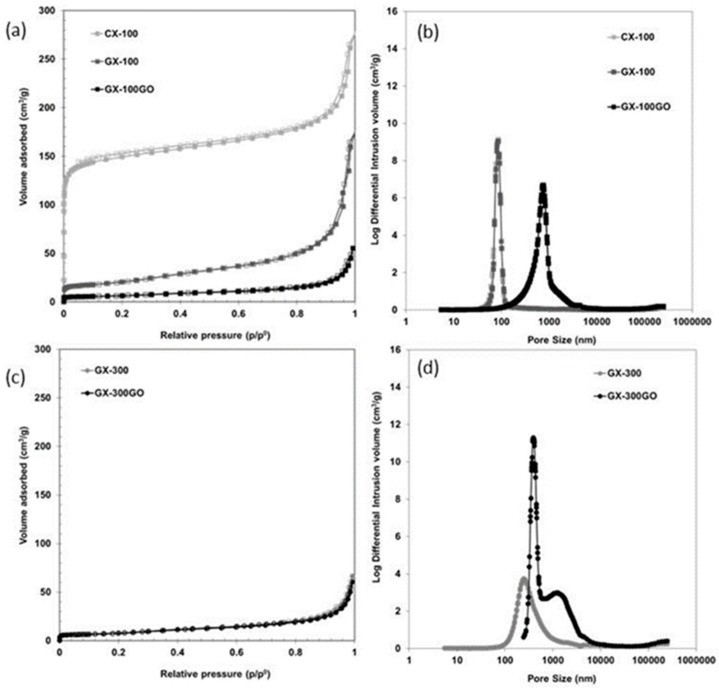
N_2_ adsorption–desorption isotherms, (**a**,**c**), and pore size distributions obtained from mercury porosimetry, (**b**,**d**), of the xerogels analyzed.

**Figure 4 materials-13-00119-f004:**
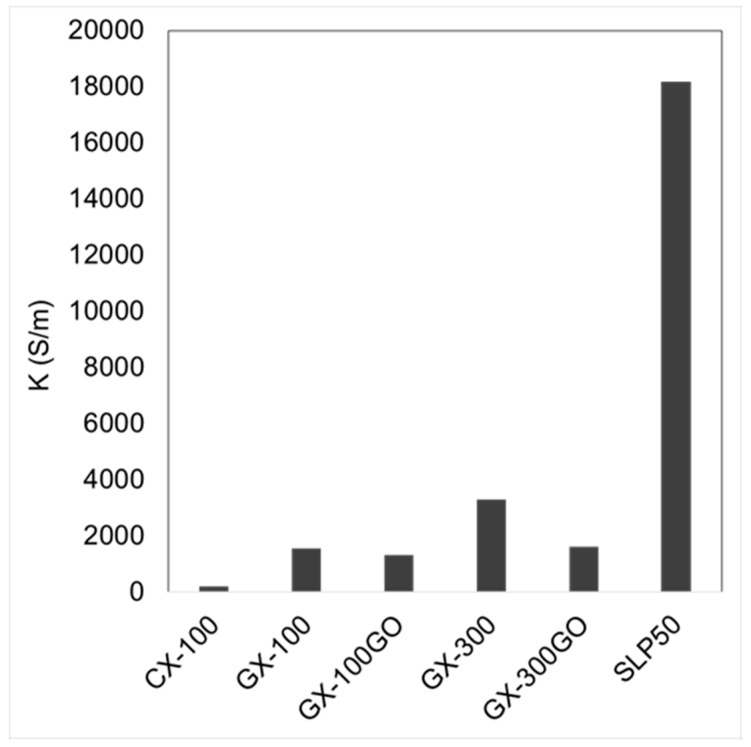
FPP conductivities of the xerogels electrodes.

**Figure 5 materials-13-00119-f005:**
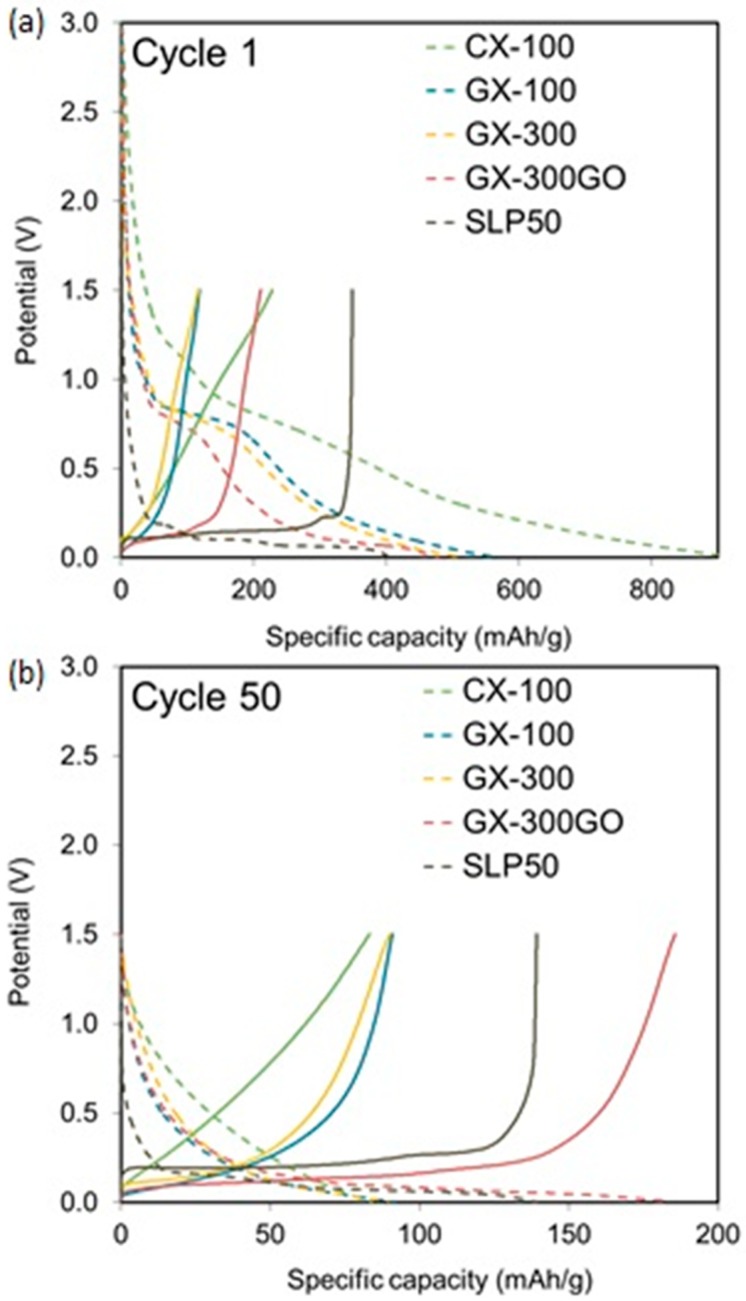
Discharge (dotted) and charge (solid) curves at a rate of C/5 for the xerogels studied during the 1st (**a**) and the 50th cycle (**b**).

**Figure 6 materials-13-00119-f006:**
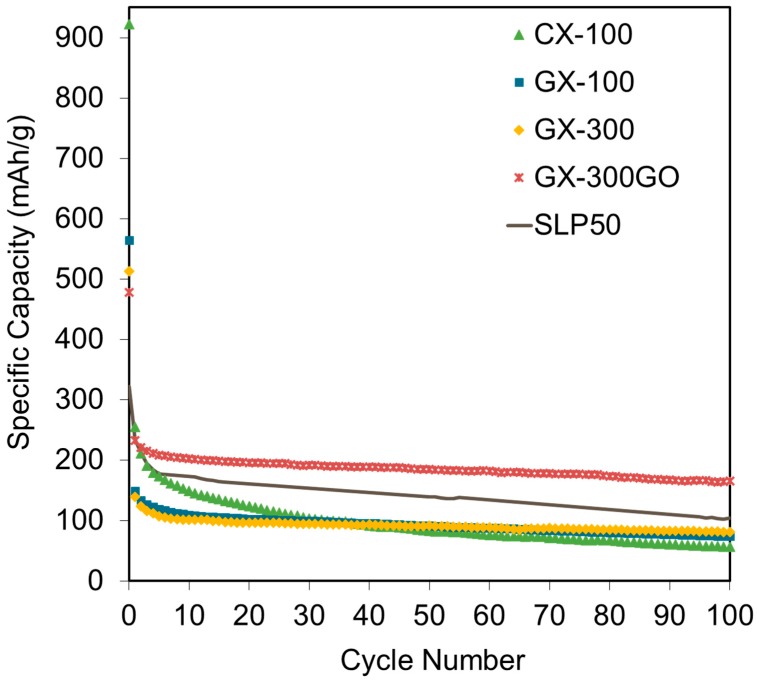
Cycling performance results for the xerogels studied and the reference material.

**Figure 7 materials-13-00119-f007:**
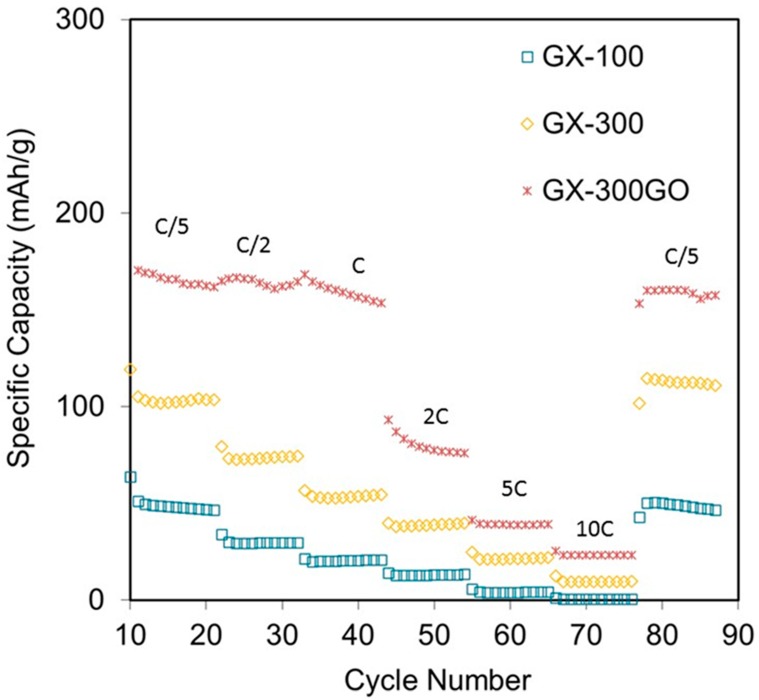
Cycling performance at different C-rates for the gels studied.

**Table 1 materials-13-00119-t001:** Structural properties of the samples.

	Graphitic		Non-Graphitic		Graphitic Contribution	S_BET_
	d_002_ (nm)	L_c_ (nm)	d_002_ (nm)	L_c_ (nm)	(%)	m^2^ g^−1^
CX-100	–	–	–	–	–	591
GX-100	0.338	14.1	0.350	2.0	7.5	71
GX-300	0.337	12.3	0.350	1.8	22.6	22
GX-100GO	0.338	12.9	0.352	1.4	32.9	29
GX-300GO	0.338	13.2	0.356	0.8	29.4	26
SLP50	0.335	53	–	–	100	5

**Table 2 materials-13-00119-t002:** Electrochemical parameters from galvanostatic cycling at a rate of C/5.

	Cirr (1stcycle)	R (50thcycle)	R (100thcycle)	Ec (1stcycle)	Ec (50thcycle)
	(%)	(%)	(%)	(%)	(%)
**CX-100**	72.4	36.8	25.2	27.6	99.2
**GX-100**	73.7	77.6	62.7	26.3	99.2
**GX-300**	72.8	78.3	69.9	27.2	100.5
**GX-300GO**	51.4	87.7	77.9	48.6	100.5
**SLP50**	11	59.6	43.8	89.0	99.9

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
