# Peer review of "Graphitized Carbon Xerogels for Lithium-Ion Batteries"

_materials, 2019, doi:10.3390/ma13010119_

Round 1

Reviewer 1 Report

This work is focused on the preparation and the performance assessments of carbon xerogel negative electrodes, which have been prepared at different levels of graphitisation for different heat treatments. Carbon xerogels with small addition of graphene oxide (0.2 wt% of GO) were also prepared and then graphitized at 1000° and 2800°C. A graphite material (TIMREX® SLP50) was used as a reference material of the electrodes.

These carbon xerogel-based graphitized hybrid materials were then characterized by several physical-chemical methods including SEM, HRTEM, N2 sorption and X-ray diffraction (XRD). Besides, specific electrical resistance measurements were performed using the four-point probe method for the different samples tested.

The manuscript is well organized and written even if it has small weaknesses that need to be revised by the authors.

A list of other specific technical comments and key points that could considered

Specific technical issues and comments

As the mass loading, porosity and thickness of the electrodes are key parameters of the electrodes of lithium-ion batteries, it is recommended to include the missing ones (i.e. the mass loading and thickness) and also the geometric areas of electrodes, as well. Authors should also report the Coulombic efficiencies with cycles, maybe in in Figure 6. The Coulombic efficiency of the first cycle and that of subsequent cycles are very important factors in determining the long-term stability and capacity retention of lithium-ion batteries.

To provide and attract a more general interest from readers, some discussion with similar works (or a comparison table) of the literature on the use of different graphites as anode electrode materials or other hard carbon materials (or even templates) could be inserted before the conclusions.

Author Response

First of all we want to thank the reviewer for your constructive revision and helpful comments. We have modified the manuscript according to your suggestions in the revised version re-submitted. Those, along with suggestions from other reviewers, are highlighted in yellow following the instructions of the Editor.

Comment 1. As the mass loading, porosity and thickness of the electrodes are key parameters of the electrodes of lithium-ion batteries, it is recommended to include the missing ones (i.e. the mass loading and thickness) and also the geometric areas of electrodes, as well.

Answer: OK. We have included the thickness of the copper disk and the active material in the experimental part in lines 159 and 161, respectively. The mass loading was also added in line 161.

Comment 2. Authors should also report the Coulombic efficiencies with cycles, maybe in in Figure 6.

Answer:  We totally agree, the Coulombic efficiency is an important parameter to evaluate the performance of a battery, therefore the coulombic efficiencies of the samples at the 1st and the 50th cycle were included but in Table 2. A brief comment was also added in line 307.

Comment 3. To provide and attract a more general interest from readers, some discussion with similar works (or a comparison table) of the literature on the use of different graphites as anode electrode materials or other hard carbon materials (or even templates) could be inserted before the conclusions.

Answer: In accordance with our experience, it is difficult to compare results with those of other authors as the equipment and the conditions at which the experiments were performed (i.e. the current collector, the electrolyte use, the assembly of the battery) are different for each study and take an important part in the final performance of the device, for that reason we characterized at our conditions a commercial carbon (SLP50) to use as reference. However, it is true that the importance of the result obtain was not enough clarified. Accordingly, we have included a short paragraph before conclusions to remark the advance performance of GX-300GO.

Reviewer 2 Report

This is a well written and interesting report on the effect of texture and structure of carbon xerogels and hybrid graphene/xerogel carbons on their performance as electrodes in lithium-ion batteries.

The manuscript can be accepted for publication after minor review, according to the following comments and suggestions.

Line 167: please explain the meaning of "C-rate" in the text.

Figure 5: Data for the reference carbon (SLP50) could/should be included.

Line 308, caption of Table: THis is Table 2, not Table 1

Line 335: Authors refer "Fig. 1 d", but there is no such figure. Perhaps they mean 3d?

Figure 7: The caption of this figure, "Rate performance..." must be changed to "Cycling performance..."

Authors should provide some explanation to why there are different values in Figures 6 and 7, for the same sample. For instance, for GX300GO, at 10 cycles, the specific capacitance is higher than 200 in Figure 6, but it is lower than 200 in figure 7.

Author Response

First of all we want to thank the reviewer for your time and helpful comments. We have modified the manuscript according to your suggestions in the revised version re-submitted. Those, along with suggestions from other reviewers, are highlighted in yellow following the instructions of the Editor.

Comment 1. Line 167: please explain the meaning of “C-rate” in the text.

Answer: OK. The C-rate meaning was introduced in the text in line 169.

Comment 2. Figure 5: data for the reference carbon (SLP50) could/should be included.

Answer: Ok. Data of the reference material have been included in Figure 5.

Comment 3. Line 308, caption of Table: This is Table 2, not Table 1.

Answer: The mistake has been corrected.

Comment 4. Line 335: Authors refer “Fig. 1d”, but there is no such figure. Perhaps they mean 3d?.

Answer: You are totally right, the mistake has been corrected.

Comment 5. Figure 7. The caption of this figure “Rate performance…” must be changed to “Cycling performance…”.

Answer: Ok. The caption of this figure was changed accordingly.

Comment 6. Authors should provide some explanation to why there are different values in Figures 6 and 7, for the same sample. For instance, for GX300GO, at 10 cycles, the specific capacitance is higher than 200 in Figure 6, but it is lower than 200 in figure 7.

Answer: You are right. Minor differences in the capacitance values of the materials are observed between figures 6 and 7. The performance of the device not only depends on the active material but also on the quality of the battery assembly, i.e. the current collector, the separator and the pressure applied during the assembly to ensure a good contact between the components. Although the same active material was used, to perform the experiments required for obtaining the results showed in figures 6 and 7 different battery assemblies were employed, for that reason small differences in the capacitance values were obtained. However, it is important to remark that the trends between samples remain analogous and so the conclusions obtained.

Reviewer 3 Report

The paper is of current interest for the electrochemistry community. It is written in a clear way. The use of carbon xeerogel with different degrees of graphitization is very interesting and deserves publication. Accordingly, I suggest to publish it almost as is. The only modification required is regarding the number of self-citations, which is too high (please consider also other works related to xerogel for other applications, see for example: (2017) Materials, 10 (9), art. no. 1092.DOI: 10.3390/ma10091092; (2014) Journal of Materials Chemistry A, 2 (33), pp. 13713-13722. DOI: 10.1039/c4ta02108h; (2014) Applied Catalysis B: Environmental, 147, pp. 947-957. DOI: 10.1016/j.apcatb.2013.10.031;(2013) ChemCatChem, 5 (12), pp. 3770-3780. DOI: 10.1002/cctc.201300542; etc.)

Author Response

First of all we want to thank the reviewer for your time and helpful comments. In fact, we have included practically all your suggestions in the new version of the manuscript. Those, along with suggestions from other reviewers, are highlighted in yellow following the instructions of the Editor.

Comment 1. ….The only modification required is regarding the number of self-citations, which is too high (please consider also other works related to xerogel for other applications, see for example: (2017) Materials, 10 (9), art. no. 1092.DOI: 10.3390/ma10091092; (2014) Journal of Materials Chemistry A, 2 (33), pp. 13713-13722. DOI: 10.1039/c4ta02108h; (2014) Applied Catalysis B: Environmental, 147, pp. 947-957. DOI: 10.1016/j.apcatb.2013.10.031;(2013) ChemCatChem, 5 (12), pp. 3770-3780. DOI: 10.1002/cctc.201300542; etc.)

Answer: OK. We have added all the references you have suggested and another two (Microporous and Mesoporous Materials, vol 275,278-287, 2019; and Int. J. Electrochem. Sci., Vol. 8, 2013) that, in our opinion, can improve the manuscript. However, we consider that in this case our self-citations are totally necessary for a better understanding of this research.
